# Characterization of a Collection of Colored Lentil Genetic Resources Using a Novel Computer Vision Approach

**DOI:** 10.3390/foods11243964

**Published:** 2022-12-07

**Authors:** Marco Del Coco, Barbara Laddomada, Giuseppe Romano, Pierluigi Carcagnì, Shiv Kumar, Marco Leo

**Affiliations:** 1Institute of Applied Sciences and Intelligent Systems (ISASI), National Research Council (CNR), Via Monteroni, 73100 Lecce, Italy; 2Institute of Sciences of Food Production (ISPA), National Research Council (CNR), Via Monteroni, 73100 Lecce, Italy; 3International Center for Agricultural Research in the Dry Areas (ICARDA), Beirut 1108-2010, Lebanon

**Keywords:** pulses, lentil grains, germplasm resources, morphological descriptors, image analysis

## Abstract

The lentil (*Lens culinaris* Medik.) is one of the major pulse crops cultivated worldwide. However, in the last decades, lentil cultivation has decreased in many areas surrounding Mediterranean countries due to low yields, new lifestyles, and changed eating habits. Thus, many landraces and local varieties have disappeared, while local farmers are the only custodians of the treasure of lentil genetic resources. Recently, the lentil has been rediscovered to meet the needs of more sustainable agriculture and food systems. Here, we proposed an image analysis approach that, besides being a rapid and non-destructive method, can characterize seed size grading and seed coat morphology. The results indicated that image analysis can give much more detailed and precise descriptions of grain size and shape characteristics than can be practically achieved by manual quality assessment. Lentil size measurements combined with seed coat descriptors and the color attributes of the grains allowed us to develop an algorithm that was able to identify 64 red lentil genotypes collected at ICARDA with an accuracy approaching 98% for seed size grading and close to 93% for the classification of seed coat morphology.

## 1. Introduction

The lentil (*Lens culinaris* Medik.) is among the most commonly cultivated grain legumes and one of the oldest domesticated crops worldwide. While Canada, India, Turkey, and the United States are the main lentil-producing areas [1], the countries surrounding the Mediterranean basin hold the largest quota of genetic variation for the species [2]. However, a drastic reduction in lentil cultivation has occurred in the last century in Mediterranean countries due to changing lifestyles and eating habits [3] and due to climate changes resulting in the loss of many landraces [4]. So far, several local farmers and international genebanks are the custodians of the existing biodiversity, including neglected landraces. A first step to guarantee their preservation is to characterize them both in situ and ex situ starting from the grain seed size, which is one of the major qualitative descriptors of lentil [5]. Besides grain size, grain color is another important morphological trait associated with quality [6,7]. In particular, grain color is given by the grain ground color (i.e., green, grey, brown, black, or pink), the cotyledon color (i.e., yellow, orange, red, or olive green), the pattern of testa (i.e., absent, dotted, spotted, marbled, or complex), and the color of pattern on testa (absent, olive, grey, brown, or black) [8].

While uniform, light-colored lentil grains are preferred in the United States and Europe [9], dark seed coats are more appreciated in Asia and in some Mediterranean countries [10]. The seed color is due to the accumulation of polyphenols in the grain tissues, which also influences plant adaptation to environmental constraints [11,12], antioxidant capacity, and other health-related benefits in humans [13].

To estimate the lentil seed size and color, several objective and non-destructive methods were proposed. Recently, some image-based methods were used for sizing lentil grains through segmentation by thresholding in hue–saturation–value (HSV) color space [14]. Color and appearance grading were analyzed by image color, color distribution, and textural features using Wilk’s lambda and the average-squared-canonical correlation as the criteria of significance [15]. Another approach was developed for discriminating between colored lentil genotypes (red/yellow) with large-, medium-, and small-sized grains [10]. Otsu’s automatic threshold algorithm was used for seed segmentation, whereas classification was carried by back propagation neural network and support vector machine classifiers fed by morphological features (eccentricity, axial length, area, and perimeter). Finally, it is worth mentioning that a portable imaging system (BELT) supported by image acquisition and analysis software (phenoSEED) was created for small seed optical analysis [16]. Based on that, phenoSEED was able to process images returning results of extracted seed shape, size, and color. However, none of the previous works provided a complete algorithmic pipeline for lentil seed analysis, including the classification of testa patterns, because they focused on the estimation of size and color cluster [17]. In addition, the current measurement systems require initial calibration procedures, well-defined acquisition setups, and/or online adjustments by operators.

In this paper, we developed a novel approach to overcome the drawbacks of the previous methods by introducing a pipeline fed by RGB (red, green, and blue) images. These were acquired in the laboratory under hybrid lighting systems with the use of available natural daylight, supplemented with artificial light to meet the required level of seed illumination. Our aim was to operate under the most common conditions in which in situ custodians of lentil germplasm work, enabling them to undertake a reliable and objective characterization of seed morphology of neglected lentil accessions. In this study, we used an off-the-shelf mobile phone camera to acquire the images of the 64 pigmented lentil genotypes comprising a collection of pigmented lentils received from the International Center for Agricultural Research in the Dry Areas (ICARDA-Lebanon). The images enabled us to recover very accurate information about lentil seed size and testa patterns exploiting a randomized circle detection strategy, local binary patterns descriptors, and XGBoost classifier, which were applied for the first time in seed morphology analysis.

## 2. Materials and Methods

The proposed approach does not require a specific acquisition setup. Images can be taken from a camera mounted on off-the-shelf mobile devices. It only requires a “quasi” top view of the seeds (to avoid huge perspective deformations), a little radiometric contrast between the seeds and the background (seeds should be clearly visible in the images), and a sheet of graph paper pictured in a small area of the acquired images (for automatically controlling the image scale). The mobile phone can be handled by the user during the acquisition or placed on a support (avoiding, this way, motion artefacts).

The acquisition image campaign was carried out under hybrid lighting systems with the use of available natural daylight, sometimes supplemented with artificial light to meet the required level of seed illumination. The focus and iris parameters of the camera were set at fixed values.

The images were acquired from a camera of a smartphone positioned on a height-adjustable holding structure (camera support) on different days. The camera support was moved between acquisition campaigns, and we found that its repositioning contained a non-negligible error for the purposes of the evaluation process here described. This means that the images in some cases showed a different scaling factor that was handled by the algorithms. In Figure 1, the acquisition setup is shown. The dataset is publicly available at https://github.com/beppe2hd/unconstrainedLentils (accessed on 10 November 2022).

### 2.1. Plant Materials

In the present study, we analyzed the grains of 64 lentil genotypes received by ICARDA in Lebanon, including 48 varieties released in 19 different countries between 1984 and 2018, 9 germplasm accessions, and 7 elite breeding lines developed at ICARDA in Lebanon (Table A1). For brevity, the genotypes were labeled using a progressive numeration from 1 to 64, while the information about each genotype is shown in Appendix A. The plant materials were grown in the 2018/19 season at Terbol, Lebanon (33.81° N, 35.98° E), at 890 m above sea level, and characterized by cool winter and high rainfall as a typical continental to semi-arid climate with clay soil.

The input data for the computer vision analysis were 64 images picturing lentil seeds randomly positioned on a black surface. In each image, the number of lentil seeds varied. In total, the seeds in the images totaled 1408.

Within each image, the seeds were separated well from each other, but in some cases, there was a superimposition among close seeds and partial occlusions occurred. The number of seeds randomly varied depending on the availability of the seed samples.

Due to the unconstrained acquisition conditions, some seeds positioned out of the image center were not pictured in focus, and the images showed different radiometric values depending on the lighting conditions at the shot event.

### 2.2. Grain Traits

The proposed algorithmic pipeline (represented in Figure 2) consisted of several steps aimed at extracting information about the image contents. In more detail, the full image was processed by the “square detector” module in charge of the identification of the seeds in the image. Detected circles were then employed on one side to extract the patches of the lentil seeds and, on the other side, to retrieve the average seeds’ radius in pixels. Other than that, the original images were also processed by the “square detector” exploiting the graph paper as a reference to detect the small squares. The detected squares were then used to compute the scaling factor, which is useful for estimating the average seeds’ radius in millimeters. In addition, the extracted patches were provided, one by one, as input to the seed texture classification module returning their texture class (rare, sparse, dense). Subsequently, a seed pattern classification module was employed to define the seed class estimation (absent, dotted, spotted, marbled, complex).

### 2.3. Seeds Segmentation

At the beginning, the problem of segmenting seeds, i.e., extracting foreground objects (the seeds) from the background had to be addressed in each input image. To exploit the curvature of edges on seeds, a circular shape detector was exploited for the scope. Circular shapes in the input images were detected using a randomized approach [18] that can accurately detect circles within a limited number of iterations, maintaining a sub-pixel accuracy even in the presence of a high level of noise and partially occluded edges. In addition, it did not require a priori knowledge about searched radius enabling the detection of circles having different sizes, even in the same image, as is usual when analyzing images picturing lentil seeds. This way, in each image, a set of circular regions was extracted, and, for each region, the center and the radius were returned.

A radiometric check was conducted to discard false circles containing large background areas (this can happen due to the randomized nature of the algorithm chosen to avoid a priori information about expected dimensions).

In the considered experiments, circular regions with more than 10% of background pixels were discarded. Moreover, a statistical check on detected radii was conducted, and values outside within two standard deviations were discarded too. To measure the average size of the seeds in the image, the estimation of the scaling factor in each image was first carried out. This allowed the system to transform the detected radii from pixel to mm. Each image contained a 1 mm graph paper pictured on top of it. The square shapes were detected by using edge extraction and contour following strategies, and the average square side in pixels was estimated. This allowed us to extract, image by image, the spatial coverage S of each pixel on the plane where the seeds were positioned and then obtain a measure in the real world for the object on that plane.

Using a formula, the scale parameter S could be computed as S = ADV/EDV, where EDV was the estimated average measure (in pixels) of the squares on the graph paper, and ADV was the a priori known measure (in mm) of the square side (in the considered graph paper, it was 1 mm).

Figure 3 shows the process on the graph paper of the image corresponding to the lentil grain sample n. 6. It is worth noting that only a portion of squares was detected because not all the sides of the squares could be retrieved, even after image enhancements. Anyway, the detected squares were enough to build reliable statistics for estimating the length of the sides.

This way, the seed measures were referred to as the measures of their orthogonal projection on the laying plane. It is worth noting that this was not a limitation because the final goal was to measure the radius of the sphere approximating a seed. The average radius in mm of seeds in each image was finally computed by multiplying the values in pixels with the computed scale factor *S*. This way, the estimated radius of the lentil species in each input image was the outcome of the considered algorithmic step.

### 2.4. Lentil Seeds Pattern Classification

According to the scientific literature, the classification for seed pattern includes 5 classes, namely, “absent”, “marbled”, “dotted”, “spotted”, and “complex” [8]. However, this classification is not satisfactory to describe the variation that many lentil genotypes show. To automatically address this problem, a two-step procedure was introduced and is represented in Figure 4.

The patches extracted around the circles detected as described in the previous section were the input of the procedure. Each patch was processed independently from the image from which it was extracted. The first step addressed a 3-classes problem by analyzing texture patterns and color histograms. Each patch was classified as containing rare, sparse, and dense texture patterns. Seeds with rare texture did not contain specific patterns, or, if some blobs were present, they were few and localized in small regions. Each cluster could have a different size and shape. Sparse textures were characterized by a uniform distribution of patterns on the surface of the seed. In other words, there was no substantial texture difference among different regions of the seed.

Finally, dense texture showed patterns covering the whole surface of the seed, and they looked like a large stain.

Some examples of rare-, sparse-, and dense-textured seeds are reported in Figure 5. In the second algorithmic step, the patches having rare texture patterns are disambiguated and definitively classified as absent or spotted, whereas the patches with sparse-textured patterns are labeled as dotted or complex. Densely textured patches were directly labeled as marbled.

The algorithmic details of the methodological steps are detailed in the following steps. In the first step, the local binary pattern (LBP) descriptors [19] were computed in each circular area extracted as described in the previous section. The length of the LBP descriptor was 18. To the end of each LBP descriptor, a 180 bins histogram was attached to collect information about each color channel (60 bins for each RGB channel). As a result, the feature vector consisted of 198 items describing the color and the texture. These vectors were used then to train a tree-based ensemble method named XGBoost [20]. Once each seed was classified as one of the three classes, a morphological region analysis was applied to determine if a rare texture corresponded to an absent or spotted subclass and if a sparse texture was relative to a dotted or complex subclass. The morphological analysis [21] aimed at separating relevant patterns (foreground) from the uniform surface of the seeds (background). The analysis consisted of several sequential steps: edge detection using the Canny operator with adaptive threshold, connectivity analysis, closing operation (dilation and erosion using the same rectangular structuring element for both operations), contour following, and, finally, the area computation of the detected regions. Among the seeds with rare texture, those with no significant regions were labeled as absent, whereas those with a few regions were labeled as spotted. On the other hand, among the seeds with sparse texture, those with large regions covering more than 50% of the surface were labeled as complex, and the remaining ones were labeled as dotted.

## 3. Results

### 3.1. Seeds Segmentation and Measuring

Using the algorithm described in the previous sections, 1549 circles were detected in the 64 images of the provided dataset. After the radius check, 1136 patches were retained, and finally, after the radiometric check, 940 were definitively retained for further processing. The seeds pictured in the 64 images totaled 1270. This means that circle detection correctly retrieved more than 74% (940 × 100/1270) of seeds that were the input of the following processing steps. No false circles were detected. The missed circles were mainly due to defocusing and occluded edges (seeds too close to each other). The circle detection algorithm had thresholds concerning the minimum number of edge pixels to detect a circle. This parameter was set to avoid false circles (highly likely in a cluttered scene such as the processed one, but, on the other side, this choice led to some missing detection that could be better tolerated considering the multiple seeds for each species). Figure 6 reports two examples of discarded circles.

In Figure 7, the outcomes of the processing of image 39 are reported. On the left, the whole image is reported to show the graph paper pictured on the top, from which it was possible to estimate the scale factor. In this example, each mm was estimated as containing about 56.64 pixels, and then the scale factor was S = 0.01765. On the right, the outcomes of the algorithm aiming at finding circular shapes are reported. In the image of lentil sample n. 20, the seeds were correctly detected (one was missed on the top-left part, which could be due to defocusing that led to the partial detection of edge points). No circle was discarded by radiometric or statistical checks since all the detected circles fitted actual seeds.

The average radius of detected circles was 146 pixels in the example, and then the size of the seed was computed as 146 × *S* = 146 × 0.01765 = 5.16 mm. The absolute error *E* was then estimated as $E=abs(r-\hat{r}$), where *r* is the actual radius of the seed, and $\hat{r}$ is the corresponding estimated value. In the example, *E* = *abs* (5 mm − 5.16 mm) = 0.16 mm.

Figure 8 reports the comparison between the measured and the estimated average size of the seeds in each of the 64 input images and the relative absolute error. The mean of the absolute errors was $\bar{E}=\frac{\sum_{i}E_{i}}{N}=0.1\;\mathrm{mm}$, where *N* = 64 is the number of input images.

The root mean square error and *R*-squared values were instead:$$RMSD=\sqrt{\frac{\sum_{i=1}^{N}{\left(x_{i}-{\hat{x}}_{i}\right)}^{2}}{N}}=33.35\;\mathrm{mm}$$
$$R^{2}=1-\frac{RSS}{TSS}=0.82$$
where *RSS* is the sum of squares of residuals, and *TSS* is the total sum of squares.

The accuracy of measurement was $cc=\sum_{i}\frac{x_{i}-\left(x_{i}-{\hat{x}}_{i}\right)}{x_{i}}=0.98$.

### 3.2. Lentil Seeds Pattern Classification

To assess the accuracy in pattern classification, all the 940 patches extracted from full images were annotated by a team of experts. The first experimental phase was devoted to validating the ability of the proposed pipeline to address the three classes of classification, i.e., to recognize the texture pattern of the seeds as rare, sparse, or dense. The method exploits LBP and histograms as features and XGBoost as the classifier. A square patch (side equal to the relative circle diameter) was extracted around each of the 940 detected circles. The patches were labeled as follows: 635 seeds with rare patterns, 25 seeds with dense patterns, and 280 seeds with sparse patterns. Considering the size of the available dataset, the leave-one-out approach was used to test the performance of the proposed method. First, the three-class problem was considered.

The results are reported as a confusion matrix in Table 1, whereas statistical scores were F1-Score = (correct classifications)/(correct classifications + wrong classifications) = 0.98. overall accuracy = correct classification/number of examples = 0.97. balanced accuracy = (correct classification class 1/number of examples class 1 + correct classification class 2/number of examples class 2 + correct classification class 3/number of examples class 3)/3 = (631/635 + 270/280 + 18/25)/3 = 0.89.

The second experimental phase dealt with the finer classification of lentil seed classes starting from the three-class coarse classification above. The 64 images used for the test contained 511 seeds classified as absent, 124 as spotted, 247 as dotted, 25 as marbled, and 33 as complex.

To assess the algorithm based on morphological region analysis, the seeds containing rare textures and sparse patterns were processed. In Figure 8, an example of a sparse textured pattern is finally classified as complex relying on morphological analysis. Seeds with rare texture patterns were classified as presented in the confusion matrix of Table 2 providing an F1-score of 0.9221 and an overall accuracy of 0.93.

Classification results concerning the sparse texture patterns were reported in the confusion matrix of Table 3 with an F1 score of 0.86 and an overall accuracy of 0.84.

In the third experimental phase, all the patches were classified according to the two-step scheme in Figure 3. This brought to the results presented in the confusion matrix of Table 4 with an overall accuracy of 0.93, a balanced accuracy of 0.98 + 0.69 + 0.96 + 0.76 + 0.94 = 0.87, and an F1-score of 0.92.

In Figure 9, some examples of correct lentil pattern classifications are reported, whereas Figure 10 reports two examples highlighting the algorithmic details of a sparse texture, finally classified as complex, and a rare texture, finally classified as spotted.

## 4. Discussion

The seed size, color, and testa appearance of lentils are important grading factors influencing the nutritional quality [7] and market acceptance [9,10]. In this study, we developed a new computer vision system for assessing the seed size, color grading, and testa morphology of lentils using an off-the-shelf mobile phone camera to acquire the images of the 64 pigmented lentil genotypes obtained from the International Center for Agricultural Research in the Dry Areas (ICARDA-Lebanon). The images enabled us to recover very accurate information about lentil seed size and testa patterns by exploiting an algorithmic pipeline leveraging a randomized circle detection strategy, local binary patterns descriptors, and XGBoost classifier, which were applied for the first time in seed morphology analysis. It is worth noting that the provided outcomes are independent of the subjectiveness and the expertise of humans, and this can make the characterization uniform in time and among different places allowing a more powerful sharing of information.

To give evidence of the scientific contribution, it is worth noting that such an automatic classification of testa patterns was never addressed before. Previous attempts only concentrated on colour and size.

For the first time, the color components of the seeds were not exploited for color grading, but they were combined with local binary features allowing the characterization of the testa patterns and their classification according to the prominent literature for lentil assessment. In addition, size measurements were not constrained by calibrated acquisition systems or seed handling facilities.

Among the existing tools for seed measurement relying only on 2D measurements, the Belt and phenoSEED platforms represent a milestone for this research field. Anyway, according to the results reported in a related paper [20], experienced very large variance even among the samples of cultivated lentils. Furthermore, the reported interquartile range for area measurements shows that the error was typically around 1 mm against the 0.1 mm reached by using the proposed algorithm pipeline.

On the other hand, there exist platforms providing better accuracy than the proposed one [14], but they were obtained by combining 2D and 3D data acquired by a calibrated line-scan camera and laser scanners, with seeds dispensed on a vibrating comb and onto a conveyor belt so that the seeds were disjointly positioned on a conveyor system. These kinds of platforms are not cost-effective, and a skilled operator, especially for registering different acquisition sources, should control them.

The proposed pipeline can work in uncontrolled conditions (lighting and seed positioning); instead, it makes use of cameras mounted on off-the-shelf mobile devices, and it does not require any calibration or additional physical infrastructure. The only requirement, purely restricted to the size estimation, is to put a piece of graph paper in the acquired image of seeds. All the algorithmic steps do not rely on fixed thresholds, and unexpected situations are self-checked and controlled. This system works in a broad range of lighting conditions, but it can check if extreme conditions arise. Dark or saturated images are discarded at the beginning by checking lightness, as well as images not in focus are immediately notified to the user when squares on the graph paper are not visible enough. As described in the previous section, several checks were also performed in each algorithmic step to reduce uncertainty in measurement and classification (e.g., only seeds whereon a contour is visible are considered, as shown in Figure 6).

## 5. Conclusions

Farmers and other custodians of in situ lentil diversity play a critical role in the sustainable management of agriculture and food security. Simplified and reliable computer vision systems can support them in the characterization and maintenance of local lentil varieties being particularly important for smallholder farmers and farming communities in rural and marginal areas. Conserving the diversity in local lentil varieties is important in avoiding crop failure under the ongoing climate changes while contributing to a healthy diet.

In summary, the present method was developed to give rise to fast and reliable computer-based systems for the automatic recognition of lentils based on the grain size, ground color, and testa texture. This can represent the algorithmic core of a mobile app that can allow any user (not skilled) to acquire seeds and to provide it with an estimated seed size and testa class, thereby allowing users to perform fast and accurate lentil species cataloguing. On the other hand, an online web service could also be implemented to provide a labeling tool accessible from all over the world by uploading images of lentils seeds.

Future works will deal with more stressful tests on more lentil genotypes and with different acquisitions per each genotype. This could lead to a systematic investigation of the possible classification bias (e.g., color and geometric artefacts) introduced by different devices. The resulting large database of images and data could finally bring deep-learning-based approaches for describing lentil morphology and differentiating between the genotypes.

## Figures and Tables

**Figure 1 foods-11-03964-f001:**
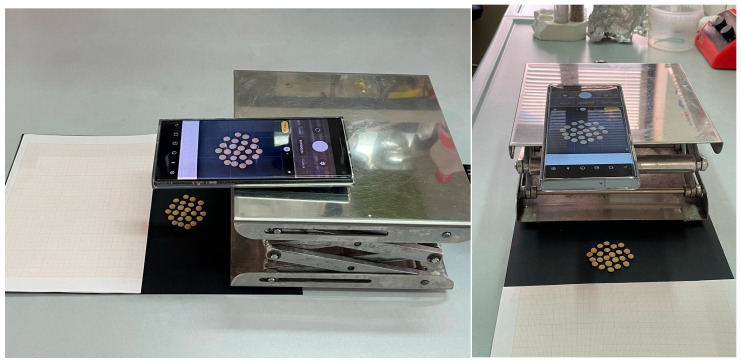
Frontal and lateral view of the setup used in the acquisition campaign.

**Figure 2 foods-11-03964-f002:**
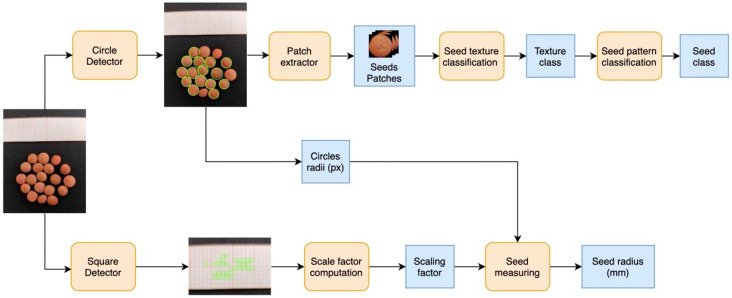
Schematic representation of the proposed pipeline performing multiple steps to define the image context.

**Figure 3 foods-11-03964-f003:**
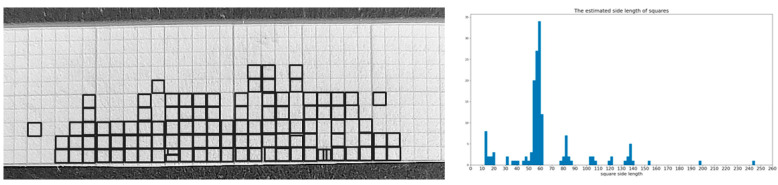
On the **left**: the graph paper and, superimposed, the detected squares. On the **right**: histogram of the side length of detected squares. The peak corresponds to the values in pixels corresponding to 1 mm.

**Figure 4 foods-11-03964-f004:**
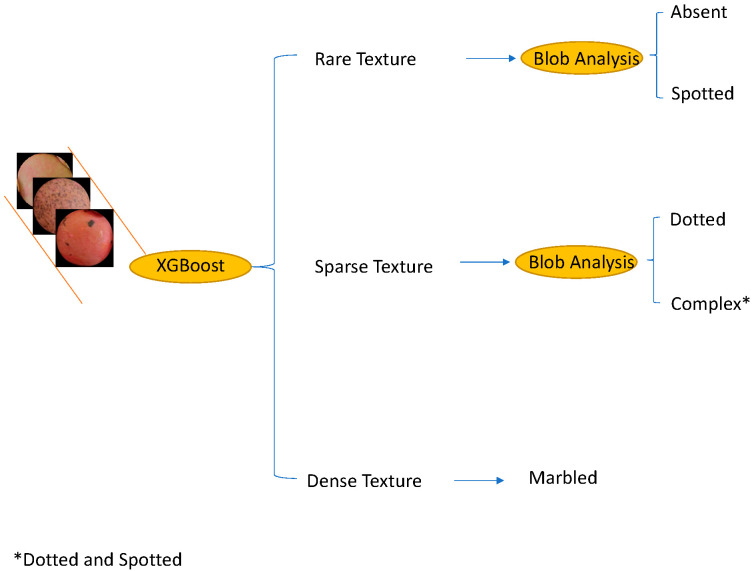
Two-step pipeline used to classify the lentil seed patterns.

**Figure 5 foods-11-03964-f005:**
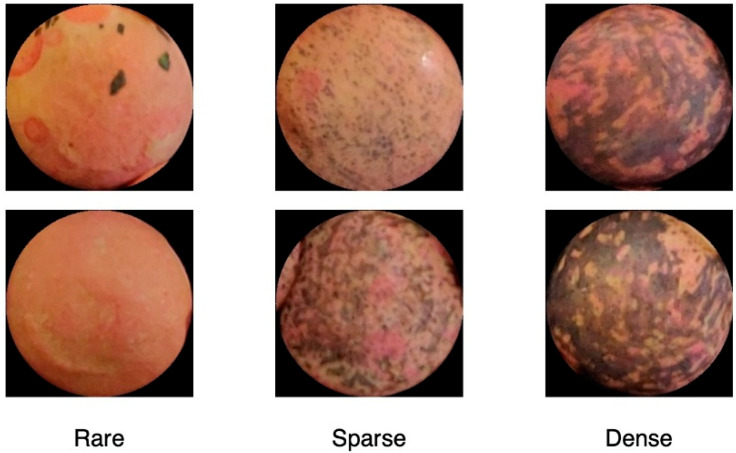
Some examples of rare-, sparse-, and dense-textured seeds.

**Figure 6 foods-11-03964-f006:**
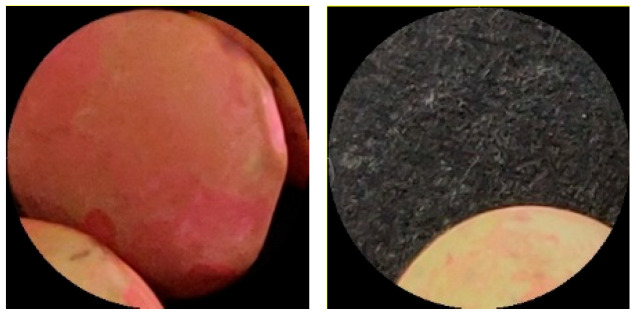
Example of circles discarded by radius (on the **left**) and radiometric (on the **right**) checks.

**Figure 7 foods-11-03964-f007:**
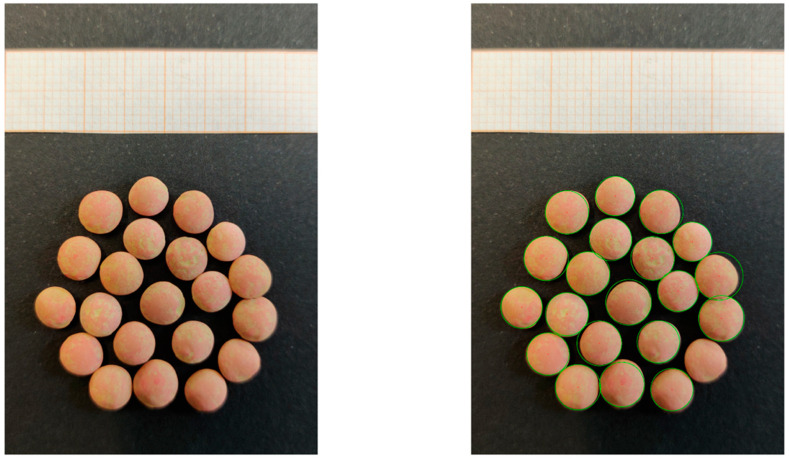
(**Left**) Input image for lentil species n. 39 and (**right**) circular shapes detected.

**Figure 8 foods-11-03964-f008:**
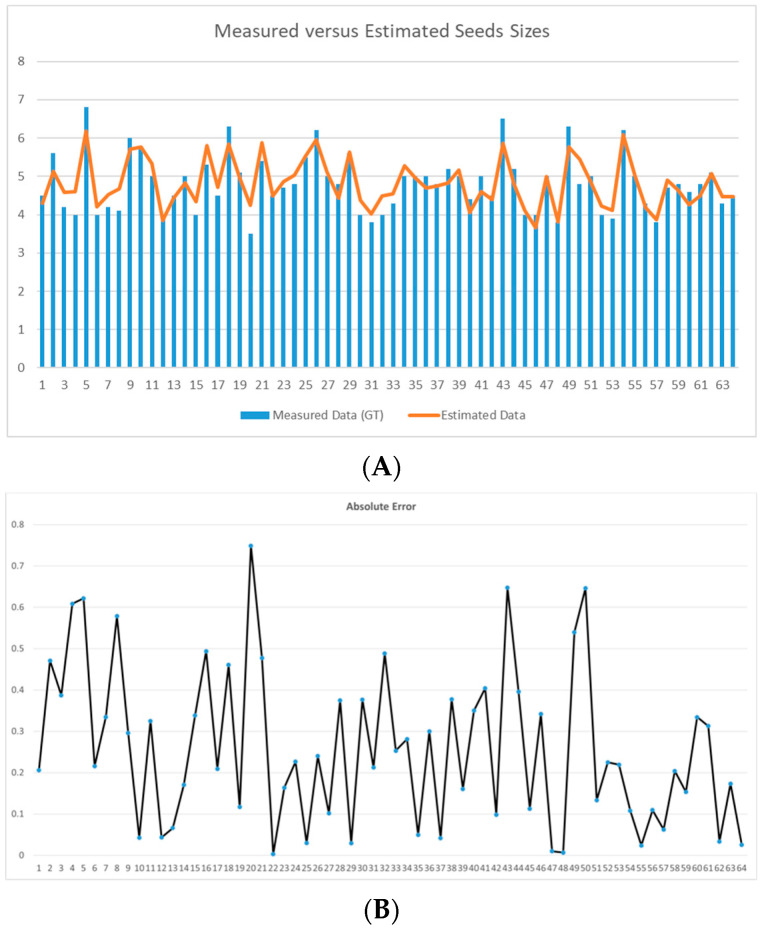
(**A**) Measured versus estimated seed sizes. (**B**) Absolute error for each processed image. The *y*-axis reports measures in mm, whereas the *x*-axis reports the corresponding input images.

**Figure 9 foods-11-03964-f009:**
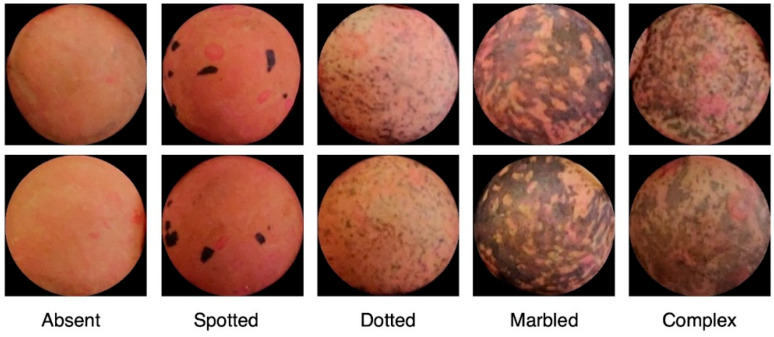
Some examples of correct lentil pattern classifications.

**Figure 10 foods-11-03964-f010:**
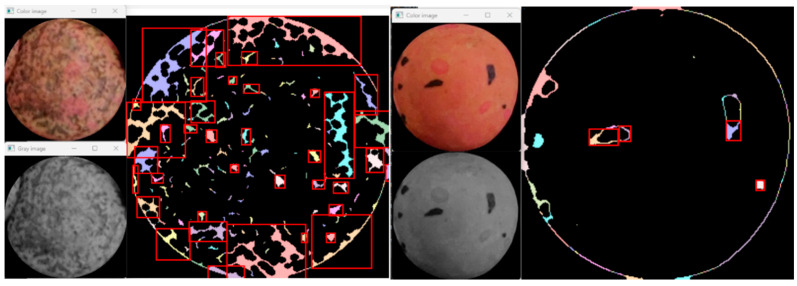
On the left, a sparse texture is finally classified as complex relying on morphological analysis. On the right, a rare texture is finally classified as spotted. Red rectangles indicate the regions automatically highlighted to make the final decision.

**Table 1 foods-11-03964-t001:** Confusion matrix for rare/sparse/dense classification. The subscript P indicates the prediction, and the subscript GT indicates the ground truth.

	Rare _P_	Sparse _P_	Dense _P_
Rare _GT_	631	4	0
Sparse _GT_	9	270	1
Dense _GT_	1	6	18

**Table 2 foods-11-03964-t002:** Confusion matrix for absent/spotted classification. The subscript P indicates the prediction, and the subscript GT indicates the ground truth.

	Absent _P_	Spotted _P_
Absent _GT_	502	9
Spotted _GT_	38	86

**Table 3 foods-11-03964-t003:** Confusion matrix for dotted/complex classification. The subscript P indicates the prediction, and the subscript GT indicates the ground truth.

	Dotted _P_	Complex _P_
Dotted _GT_	215	32
Complex _GT_	12	21

**Table 4 foods-11-03964-t004:** Confusion matrix for the overall classification. The subscript P indicates the prediction, and the subscript GT indicates the ground truth.

	Absent _P_	Spotted _P_	Dotted _P_	Marbled _P_	Complex _P_
Absent _GT_	500	9	2	0	0
Spotted _GT_	38	85	1	0	0
Dotted _GT_	0	9	236	0	2
Marbled _GT_	0	1	1	19	4
Complex _GT_	0	0	0	2	31

## Data Availability

The data used to support the findings of this study can be made available by the corresponding author upon request.

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
