# Peer review of "Characterization of a Collection of Colored Lentil Genetic Resources Using a Novel Computer Vision Approach"

_foods, 2022, doi:10.3390/foods11243964_

Round 1

Reviewer 1 Report

Authors propose a complete pipeline for lentil seed analysis and classification using a novel computer vision based approach. This is an interesting work and may attract interest from researchers in this domain. In particular, lentil seed size measurement and morphological characteristics classification are applied using a series of computer vision methods. For the computer vision part, randomized circle detection strategy, local binary pattern descriptors and XGBoost classifier are used. Although the applied computer vision techniques are well known, the application area is novel.

Authors need to explain the following minor points in order to make the flow and presentation of the paper more clear.

-     -Add a flow chart/block diagram for the complete pipeline for lentil seed analysis that uses computer vision techniques. In this way, input to different computer vision methods, classifiers/algorithms and outcome will be more clear.

-    -Show more example images for different lentil seed categories (rare, sparse, dense, absent, spotted, dotted, marbled and complex).

-    - Indicate if the dataset is publicly available or not.

Author Response

Point-by-point reply to reviewer I notes.

All the changes have been highlighted in yellow in the revised manuscript. As suggested, we also corrected minor English spelling mistakes along the manuscript.

Reviewer 1:

-     -Add a flow chart/block diagram for the complete pipeline for lentil seed analysis that uses computer vision techniques. In this way, input to different computer vision methods, classifiers/algorithms and outcome will be more clear.

Figure 2 has been modified accordingly: input to different computer vision methods, classifiers/algorithms and outcomes have been added to make the pipeline clearer and easier to understand at a glance. Moreover, the pipeline description has been improved providing more details.

-    Show more example images for different lentil seed categories (rare, sparse, dense, absent, spotted, dotted, marbled and complex).

Additional example images have been added both in Subsection 2.4 and in Section 3.

-    - Indicate if the dataset is publicly available or not.

As mentioned in the manuscript the lentil seeds are available at the International Center for Agricultural Research in the Dry Areas (ICARDA-Lebanon) and their description is available in  appendix.

Acquired images and ground truth data are available at https://github.com/beppe2hd/unconstrainedLentils (this has been added to the manuscript).

Reviewer 2 Report

The specific article describes an image analysis methodology for characterizing lentil seeds based on their size and seed coat morphology. The introduction provides sufficient background and includes all relevant references about lentil seed phenotyping while similar research has been conducted in other legume species that could be integrated. Although similar approaches have been investigated on the past, this research exploits simpler instruments for image acquisition (just a smartphone) and the overall lighting and environmental conditions are also common. This could lead to an easily adopted approach that could be applied and used for the automated classification of lentil seeds and maybe other seeds as well. A concern about the applied methodology is whether the variation in lighting and the camera movement between measurements could influence the accuracy of color discrimination and pattern measurements. Maybe a standard color marker should be included in the picture. The results of the experiment are clearly described and discussed.

Author Response

Point-by-point reply to reviewers' notes.

All the changes have been highlighted in yellow in the revised manuscript. As suggested, we also corrected some minor spell mistakes.

Reviewer 2:

A concern about the applied methodology is whether the variation in lighting and the camera movement between measurements could influence the accuracy of color discrimination and pattern measurements. Maybe a standard color marker should be included in the picture.

The camera is supposed (quasi) static but, as discussed in section 5,  in case of blur effect or other artefacts due to motion or defocusing, acquired images would be discarded by the earlier processing steps (e.g. no valid circles will be detected).

Colour components are combined with gradient ones (which do not depend on lighting conditions) and the training processes were carried out under unconstrained lighting conditions allowing classifiers to generalize knowledge in order to tackle reasonable acquisition conditions variations.   

Anyway, future works could deal with a deeper study of the impact of the acquisition conditions on classification accuracy. To achieve that, more lentil genotypes could be considered accompanied by different acquisitions per each genotypes. Some comments on that were added to the conclusion section (lines 396-400).